# Partial Recovery of Coherence Loss in Coherence-Assisted Transformation

**DOI:** 10.3390/e25101375

**Published:** 2023-09-24

**Authors:** Zhaobing Fan, Zewen Shan, Haitao Ma

**Affiliations:** School of Mathematical Sciences, Harbin Engineering University, Harbin 150001, China; fanzhaobing@hrbeu.edu.cn (Z.F.); shanzw@hrbeu.edu.cn (Z.S.)

**Keywords:** coherence, coherence transformation, coherence loss, coherence recovery

## Abstract

Coherence-assisted transformation under incoherent operations is discussed. For transformation from the pure state to the mixed state, we show that the coherence loss can be partially recovered by adding auxiliary coherent states. First, we discuss the coherence-assisted transformation for qubit states and give the sufficient and necessary condition for the partial recovery of coherence loss, and the maximum of the recovery of coherence loss is also studied in this case. Second, the maximally coherent state can be obtained in the above recovery scheme, so we give the full characterization of obtaining the maximally coherent state in a qubit system. Finally, we show that the coherence-assisted transformation for arbitrary finite-dimensional main coherent states and low-dimensional auxiliary coherent states is always possible, and the coherence loss also can be partially recovered in these cases.

## 1. Introduction

Quantum resource theory originally comes from the resource theory of quantum entanglement [1,2,3,4,5,6]. With the further evolution of quantum information, it has been gradually extended to other quantum resource theories such as coherence [7,8,9,10,11], asymmetry [12,13], reference frames [14,15], thermodynamics [16,17] and so on. In this paper, we focus on the resource theory of quantum coherence. Quantum coherence is not only a basic concept in quantum mechanics but also an important physical resource. It plays an important role in quantum information processing [18,19,20], quantum biology [21,22], quantum thermodynamics, quantum cryptography [23,24,25,26] and quantum measurement [27,28]. Therefore, it is necessary to study the resource theory of quantum coherence.

The resource theory of quantum coherence was first proposed by Baumgratz et al., and they established a rigorous framework of quantifying coherence [29]. In the resource theory of quantum coherence, due to the interaction with the environment, the decoherence phenomenon occurs. Singh et al. proved that quantum chaos and diminishing of information about the mixed initial state favors the generation of quantum coherence through unitary evolution [30]. Kurashvili et al. proved that nonunitary evolution leads to the generation of quantum coherence in some cases [31]. From another point of view, according to the golden rule of quantum resource theory, the coherence of the state does not increase under free operations. Therefore, in this paper, we study the partial recovery of coherence loss in state transformations under free operations. So far, the state transformation problem for two pure coherent states has been studied extensively, Du et al. proposed the sufficient and necessary condition for the transformation from a pure coherent state to another pure coherent state under incoherent operations [32]. Thus, we can determine whether the above state transformation can be realized. Suppose the above state transformation can be realized; it follows that the coherence of the state does not increase under incoherent operations: that is, the coherence loss in the state transformation is inevitable. For this case, Xing proposed a recovery scheme that adds an auxiliary system to the original system such that the coherence loss can be partially recovered in pure state transformations [33]. The process of adding auxiliary systems to the original system and performing a joint incoherent operation on the two systems is called coherence-assisted transformation. When the coherence of the auxiliary coherent state increases, the whole process recovers from coherence loss. However, due to the existence of noise, most states are mixed states in practice. Therefore, in this article, we discuss the recovery of coherence loss from a pure state to a mixed state.

In this paper, we discuss coherence-assisted transformation under incoherent operations. For comparable main coherent states, i.e., a pure state that can be transformed to a mixed state under incoherent operations, we show that the coherence loss can be partially recovered by adding auxiliary coherent states. First, we consider the simplest coherence-assisted transformation, i.e., both the main coherent states and auxiliary coherent states are qubit states. We give the sufficient and necessary condition for the coherence loss that can be partially recovered in a coherence-assisted transformation. Thus, we can find all of the auxiliary coherent states that satisfy the above conditions. Moreover, for given main coherent states satisfying the condition in Proposition 1, we give the concrete auxiliary coherent state that can obtain the maximum of the recovery of coherence loss. Second, as a direct application of the above recovery scheme, we give the full characterization of obtaining a maximally coherent state in a qubit system. Finally, we show that if the arbitrary finite-dimensional main coherent states satisfy a strictly majorization relation, there exist two-dimensional auxiliary coherent states that can realize the above recovery scheme.

The paper is organized as follows. In Section 2, we introduce the preliminary knowledge of quantum coherence, including incoherent states, incoherent operations, the relative entropy of coherence and the necessary and sufficient condition for the transformation from a pure coherent state to a mixed coherent state under incoherent operations. In Section 3, we discuss the coherence-assisted transformation for qubit states and obtain some interesting conclusions. In Section 4, we give the full characterization for obtaining a two-dimensional maximally coherent state in the above recovery scheme. In Section 5, we show that the recovery for arbitrary finite-dimensional main coherent states and low-dimensional auxiliary coherent states is always possible.

## 2. Preliminary

In the resource theory of quantum coherence, we first need to understand incoherent states and incoherent operations [29]. Let ${\left\{|i\rangle \right\}}_{i=1}^{d}$ be a fixed orthonormal basis in d-dimensional Hilbert space; if the density matrix is diagonal in the basis, then the diagonal density matrix is called an incoherent state. The set of all incoherent states is denoted by *I* for any δ∈I, which can be written as $\delta =\sum_{i=1}^{d}\delta_{i}\left|i\rangle \langle i\right|$. Otherwise, it is called a coherent state. Incoherent operations (IOs) are defined as the set of completely positive and trace-preserving maps for which the Kraus operators $\left\{K_{l}\right\}$ take incoherent states to incoherent states, i.e., $K_{l}\rho K_{l}^{†}/tr\left(K_{l}\rho K_{l}^{†}\right)\in I$ for all ρ∈I, where $\sum_{l}^{}K_{l}^{†}K_{l}=I$.

In order to quantify the coherence of a quantum state, we need to choose a proper coherence measure. A proper coherence measure needs to satisfy four conditions [29], and one of the conditions is monotonicity, i.e., the coherence of the quantum state does not increase under incoherent operations. Here, we adopt the relative entropy of coherence to explain the corresponding results because it is easy to calculate. Notice that the results in the paper are also equally applicable to other proper coherence measures. The relative entropy of coherence is equal to the distillable coherence [34] and can be interpreted as the minimal amount of noise required for fully decohering a state [35]; it is defined as $C_{r}\left(\rho \right)=S\left(\rho_{\mathrm{diag}}\right)-S\left(\rho \right)$, where $S\left(\rho \right)=-tr\left(\rho \log\rho \right)$ is the von Neumann entropy of the quantum state, and $\rho_{\mathrm{diag}}$ is the diagonal part of ρ. If the non-zero eigenvalues of ρ are ${\left\{\lambda_{x}\right\}}_{x=1}^{r}$, $r=rank\left(\rho \right)$, the von Neumann entropy of the quantum state can be written as $S\left(\rho \right)=-\sum_{x=1}^{r}\lambda_{x}\log\lambda_{x}$. Moreover, the coherence measure of the pure state is easy to calculate. Notice that if ρ is a pure state, $C_{r}\left(\rho \right)=S\left(\rho_{\mathrm{diag}}\right)$.

In this paper, for any pure coherent states $|\psi \rangle =\sum_{i=1}^{d}\sqrt{\alpha_{i}}|i\rangle$ and $|\phi \rangle =\sum_{i=1}^{d}\sqrt{\beta_{i}}|i\rangle$, without loss of generality, the squared coefficients $\left\{\alpha_{i}\right\}$, $\left\{\beta_{i}\right\}$ are real numbers and are arranged in non-increasing order, i.e., $\alpha_{1}⩾\alpha_{2}⩾\ldots ⩾\alpha_{d}⩾0$ and $\beta_{1}⩾\beta_{2}⩾\ldots ⩾\beta_{d}⩾0$. Let $\lambda_{\psi }=\left(\alpha_{1},\ldots ,\alpha_{d}\right)$, $\lambda_{\phi }=\left(\beta_{1},\ldots ,\beta_{d}\right)$; we say $\lambda_{\psi }$ is majorized by $\lambda_{\phi }$, i.e., $\lambda_{\psi }≺\lambda_{\phi }$, if $\sum_{i=1}^{l}\alpha_{i}⩽\sum_{i=1}^{l}\beta_{i}$ for all l=1,…,d and $\sum_{i=1}^{d}\alpha_{i}=\sum_{i=1}^{d}\beta_{i}=1$.

So far, state transformations under incoherent operations have been studied extensively, Du et al. give the sufficient and necessary condition for the transformation from a pure coherent state to another pure coherent state under incoherent operations.

**Lemma 1** ([32])**.**
*For any two pure coherent states $|\psi \rangle =\sum_{i=1}^{d}\sqrt{\alpha_{i}}|i\rangle$ and $|\phi \rangle =\sum_{i=1}^{d}\sqrt{\beta_{i}}|i\rangle$, $|\psi \rangle \overset{IO}{\to }|\phi \rangle$ if and only if $\lambda_{\psi }≺\lambda_{\phi }$.*

Furthermore, Du et al. also give the sufficient and necessary condition for the transformation from a pure state to a mixed state under incoherent operations. Here, we state the following result by majorization relation.

**Lemma 2** ([36])**.**
*For any pure state ψ=|ψ〉〈ψ| and mixed state σ, $\psi \overset{IO}{\to }\sigma$ if and only if there exists a pure state ensemble ${\left\{q_{j},|\phi_{j}\rangle \right\}}_{j=1}^{r}$ of σ such that $\lambda_{\psi }≺\sum_{j=1}^{r}q_{j}\lambda_{\phi_{j}}$.*

By Lemma 2, we can judge whether a pure state can be transformed to a given mixed state by majorization relation. For example, let $\psi =|\psi \rangle \langle \psi |,\sigma =\frac{9}{10}|\phi_{1}\rangle \langle \phi_{1}|+\frac{1}{10}|\phi_{2}\rangle \langle \phi_{2}|$, where $|\psi \rangle =\sqrt{0.4}|0\rangle +\sqrt{0.35}|1\rangle +\sqrt{0.25}\left|2\rangle ,\right|\phi_{1}\rangle =\sqrt{0.5}|0\rangle +\sqrt{0.25}|1\rangle +\sqrt{0.25}\left|2\rangle ,\right|\phi_{2}\rangle =\sqrt{0.5}|0\rangle +\sqrt{0.3}|1\rangle +\sqrt{0.2}|2\rangle .$ The above states satisfy the majorization relation $\lambda_{\psi }≺\sum_{j=1}^{2}q_{j}\lambda_{\phi_{j}}$, i.e., $\left(0.4,0.35,0.25\right)≺\left(0.5,0.255,0.245\right)$, so we can obtain $\psi \overset{IO}{\to }\sigma .$

## 3. Coherence-Assisted Transformation for Qubit States

In this part, we show that the coherence loss from a pure state to a mixed state can be partially recovered in coherence-assisted transformation. Coherence-assisted transformation [33] is the process of adding an auxiliary system to the ordinary coherence transformation. More specifically, during the state transformation for comparable main coherent states, we add auxiliary coherent states such that the whole transformation can still be realized under joint incoherent operations. Here, the initial and final auxiliary coherent states are different; such a transformation is called a coherence-assisted transformation. In a coherence-assisted transformation, when the coherence of the auxiliary coherent state increases, the coherence loss can be partially recovered. The following is a detailed description of the recovery scheme.

Suppose a pure coherent state ψ can be transformed to a mixed coherent state σ under incoherent operations, i.e., $\psi \overset{IO}{\to }\sigma$. We add a pure auxiliary coherent state $\omega_{1}$ and perform a joint incoherent operation on the two particles ψ and $\omega_{1}$ such that the coherence-assisted transformation $\left(\psi ⊗\omega_{1}\overset{IO}{\to }\sigma ⊗\omega_{2}\right)$ can still be realized, where $C_{r}\left(\omega_{2}\right)>C_{r}\left(\omega_{1}\right)$. Then, the coherence loss can be partially recovered, and the recovered coherence loss is $\Delta =C_{r}\left(\omega_{2}\right)-C_{r}\left(\omega_{1}\right)$.

We can see that the coherence of the auxiliary coherent state is increased—that is, the reduced coherence of the initial main coherent state can be partially transformed to the final auxiliary coherent state—then, the coherence loss can be partially recovered, and the recovered coherence loss is $\Delta =C_{r}\left(\omega_{2}\right)-C_{r}\left(\omega_{1}\right)$. This is due to the fact that the relative entropy of coherence satisfies additivity, i.e, $C_{r}\left(\rho ⊗\sigma \right)=C_{r}\left(\rho \right)+C_{r}\left(\sigma \right)$ [34].

The core of the coherence-assisted transformation $\psi ⊗\omega_{1}\overset{IO}{\to }\sigma ⊗\omega_{2}$ is that we need to perform joint incoherent operations. In the following, we give the specific incoherent operations that are implemented. For the main coherent states ψ=|ψ〉〈ψ| and $\sigma =\sum_{l=1}^{m}q_{l}|\phi_{l}\rangle \langle \phi_{l}|$ of dimension $d_{1}$, auxiliary coherent states $\omega_{1}=|\omega_{1}\rangle \langle \omega_{1}|$ and $\omega_{2}=|\omega_{2}\rangle \langle \omega_{2}|$ of dimension $d_{2}$, where $|\psi \rangle =\sum_{i=1}^{d_{1}}\sqrt{\alpha_{i}}\left|i\rangle ,\right|\phi_{l}\rangle =\sum_{i=1}^{d_{1}}\sqrt{\beta_{li}}|i\rangle$. According to Lemma 2, we can obtain $\psi ⊗\omega_{1}\overset{IO}{\to }\sigma ⊗\omega_{2}⟺\lambda_{\psi ⊗\omega_{1}}≺\sum_{l=1}^{m}q_{l}\lambda_{\phi_{l}⊗\omega_{2}}$; the proof of the result is similar to that found in the literature [36]. First, according to the majorization relation satisfied on the right, we define an intermediate pure state
$$|\eta \rangle =\sum_{i=1}^{d_{1}}\sqrt{\sum_{l=1}^{m}{\left|\sqrt{q_{l}}\beta_{li}\right|}^{2}}|i\rangle =\sum_{i=1}^{d_{1}}\eta_{i}|i\rangle ;$$
then, there exists an incoherent operation $\Phi_{1}$ such that $\Phi_{1}\left(\psi ⊗\omega_{1}\right)=\eta ⊗\omega_{2}$. Next, for any 1⩽l⩽m, define
$$K_{l}=\sum_{i=1}^{d_{1}}\sum_{j=1}^{d_{2}}\frac{\sqrt{q_{l}}\beta_{lj}}{\eta_{i}}\left|ij\rangle \langle ij\right|.$$

It is easy to check that the map $\Phi_{2}\left(\cdot \right)=\sum_{l=1}^{m}K_{l}^{}\left(\cdot \right)K_{l}^{†}$ is an incoherent operation, and we have $K_{l}\left(\left|\eta \rangle ⊗\right|\omega_{2}\rangle \right)=\sqrt{q_{l}}|\phi_{l}\rangle ⊗|\omega_{2}\rangle .$ Last, it is obvious that the composition of any two incoherent operations is still an incoherent operation, so $\Phi =\Phi_{2}\circ \Phi_{1}$ is an incoherent operation, and
$$\Phi_{2}\circ \Phi_{1}\left(\left|\psi \rangle \langle \psi |⊗\right|\omega_{1}\rangle \langle \omega_{1}|\right)=\sum_{l=1}^{m}q_{l}|\phi_{l}\rangle \langle \phi_{l}\left|⊗\right|\omega_{2}\rangle \langle \omega_{2}|=\sigma ⊗\omega_{2}.$$

Let us give a concrete example: the following example makes this phenomenon of partial recovery of coherence loss in a coherence-assisted transformation more intuitive.

**Example 1.** 
*Consider the states with the following form*

$$\psi =|\psi \rangle \langle \psi |,\sigma =\frac{2}{5}|\phi_{1}\rangle \langle \phi_{1}|+\frac{3}{5}|\phi_{2}\rangle \langle \phi_{2}|,$$

*where $|\psi \rangle =\sqrt{0.63}|0\rangle +\sqrt{0.37}\left|1\rangle ,\right|\phi_{1}\rangle =\sqrt{0.6}|0\rangle +\sqrt{0.4}\left|1\rangle ,\right|\phi_{2}\rangle =\sqrt{0.8}|0\rangle +\sqrt{0.2}|1\rangle .$ It is easy to see that the squared coefficients of the above states satisfy the majorization relation $\lambda_{\psi }≺\sum_{j=1}^{2}q_{j}\lambda_{\phi_{j}}$, i.e., $\left(0.63,0.37\right)≺\left(0.72,0.28\right)$, so we can obtain $\psi \overset{IO}{\to }\sigma$. At the same time, there exist auxiliary coherent states $|\omega_{1}\rangle =\sqrt{0.64}|0\rangle +\sqrt{0.36}\left|1\rangle ,\right|\omega_{2}\rangle =\sqrt{0.58}|0\rangle +\sqrt{0.42}|1\rangle$ that satisfy majorization relation $\lambda_{\psi ⊗\omega_{1}}≺\sum_{j=1}^{2}q_{j}\lambda_{\phi_{j}⊗\omega_{2}},$ i.e., $\left(0.4032,0.2368,0.2268,0.1332\right)≺\left(0.4176,0.3024,0.1624,0.1176\right)$, so we can obtain $\psi ⊗\omega_{1}\overset{IO}{\to }\sigma ⊗\omega_{2}$ and $C_{r}\left(\omega_{2}\right)\approx 0.98>C_{r}\left(\omega_{1}\right)\approx 0.94$. Then, the recovered coherence loss is $\Delta =C_{r}\left(\omega_{2}\right)-C_{r}\left(\omega_{1}\right)\approx 0.98-0.94=0.04$.*


In above coherence-assisted transformation, let $\omega_{1}$ be a given auxiliary coherent state; we can see that the choice of auxiliary coherent state $\omega_{2}$ is not unique. As in Example 1, there exists another auxiliary coherent state $|\omega_{2}\rangle =\sqrt{0.62}|0\rangle +\sqrt{0.38}|1\rangle$ such that $\psi ⊗\omega_{1}\overset{IO}{\to }\sigma ⊗\omega_{2}$ and $C_{r}\left(\omega_{2}\right)>C_{r}\left(\omega_{1}\right)$. A natural question is how to find all auxiliary coherent states $\omega_{2}$, i.e., for given states $\left(\psi ,\sigma ,\omega_{1}\right)$ that satisfy $\psi \overset{IO}{\to }\sigma$, what kind of $\omega_{2}$ can realize the coherence-assisted transformation $\left(\psi ⊗\omega_{1}\overset{IO}{\to }\sigma ⊗\omega_{2}\right)$ and $C_{r}\left(\omega_{2}\right)>C_{r}\left(\omega_{1}\right)$. Let us start with the simplest case, in which the main coherent states and the auxiliary coherent states are both qubit states $\left(d=2\right)$. Let $\sigma =q|\phi_{1}\rangle \langle \phi_{1}|+\left(1-q\right)|\phi_{2}\rangle \langle \phi_{2}|$ be a pure state decomposition of σ; notice that we only consider mixed state σ of rank-2 in this part. Then, the main coherent states ψ=|ψ〉〈ψ| and σ have the following pure state decomposition
(1)$$\begin{array}{cc} \; & |\psi \rangle =\sqrt{\alpha }|0\rangle +\sqrt{1-\alpha }|1\rangle ,\\ \; & |\phi_{1}\rangle =\sqrt{\beta_{1}}|0\rangle +\sqrt{1-\beta_{1}}|1\rangle ,\\ \; & |\phi_{2}\rangle =\sqrt{\beta_{2}}|0\rangle +\sqrt{1-\beta_{2}}|1\rangle . \end{array}$$
Here the squared coefficients are arranged in non-increasing order, which mean $\frac{1}{2}⩽\alpha ⩽1$, $\frac{1}{2}⩽\beta_{1}⩽1$ and $\frac{1}{2}⩽\beta_{2}⩽1$. In order to give our main result, we first give the following lemma.

**Lemma 3** ([37])**.**
*Suppose $A=\left\{a_{1},\ldots ,a_{n}\right\},B=\left\{b_{1},\ldots ,b_{n}\right\}$; sort the elements in B in decreasing order and denote its elements by $b^{\left(1\right)}⩾b^{\left(2\right)}⩾\ldots ⩾b^{\left(n\right)}$; then, A≺B if and only if*
$$\max_{A^{{}^{\prime }}\subseteq A,\left|A^{{}^{\prime }}\right|=l}\sum_{a_{i}\in A^{{}^{\prime }}}^{}a_{i}⩽\sum_{i=1}^{l}b^{\left(i\right)},1⩽l⩽n,$$
*and equality holds when l=n.*


**Proposition 1.** 
*Suppose $\psi \overset{IO}{\to }\sigma$; there exist auxiliary coherent states $|\omega_{1}\rangle =\sqrt{c}|0\rangle +\sqrt{1-c}|1\rangle$, $|\omega_{2}\rangle =\sqrt{d}|0\rangle +\sqrt{1-d}|1\rangle$ such that $\psi ⊗\omega_{1}\overset{IO}{\to }\sigma ⊗\omega_{2}$ and $C_{r}\left(\omega_{2}\right)>C_{r}\left(\omega_{1}\right)$, if and only if $\frac{1}{2}⩽\alpha ⩽\beta ⩽1$ and*
(1)
*when $\frac{1}{2}<c⩽\beta ,\max\left\{\frac{1}{2},\frac{\alpha c}{\beta }\right\}⩽d<c;$*
(2)
*when $\beta <c<\max\left\{\beta_{1},\beta_{2}\right\},\max\left\{\;\;\frac{\left(1-\alpha \right)c+\alpha -\beta }{1-\beta },\min\left\{\frac{c-\left(1-q\right)\beta_{2}}{q},\frac{c-q\beta_{1}}{1-q}\right\}\right\}⩽d<c$,*


*where $\beta =q\beta_{1}+\left(1-q\right)\beta_{2}$.*


**Proof.** The condition $\psi \overset{IO}{\to }\sigma$ means
(2)$$\frac{1}{2}⩽\alpha ⩽q\beta_{1}+\left(1-q\right)\beta_{2}=\beta ⩽1.$$
The condition $C_{r}\left(\omega_{2}\right)>C_{r}\left(\omega_{1}\right)$ is equivalent to
(3)$$\frac{1}{2}⩽d<c⩽1.$$
Because $C_{r}\left(\omega_{1}\right)⩽C_{r}\left(\omega_{2}\right)$ is equivalent to $\lambda_{\omega_{2}}≺\lambda_{\omega_{1}}$ in a qubit system, the equality holds only when d=c. By Lemma 2, the proof of the proposition can be reduced to finding the conditions that satisfy majorization relation $\lambda_{\psi ⊗\omega_{1}}≺\sum_{j=1}^{2}q_{j}\lambda_{\phi_{j}⊗\omega_{2}}$ and Equation (Equation 3) with assumption (2).First, we can obtain the squared coefficients of $\left|\psi \rangle ⊗\right|\omega_{1}\rangle$, $|\phi_{1}\rangle ⊗|\omega_{2}\rangle$ and $|\phi_{2}\rangle ⊗|\omega_{2}\rangle$:
(4)$$\begin{array}{cc} \; & A=\left\{\alpha c,\alpha \left(1-c\right),\left(1-\alpha \right)c,\left(1-\alpha \right)\left(1-c\right)\right\},\\ \; & B_{1}=\left\{\beta_{1}d,\beta_{1}\left(1-d\right),\left(1-\beta_{1}\right)d,\left(1-\beta_{1}\right)\left(1-d\right)\right\},\\ \; & B_{2}=\left\{\beta_{2}d,\beta_{2}\left(1-d\right),\left(1-\beta_{2}\right)d,\left(1-\beta_{2}\right)\left(1-d\right)\right\}. \end{array}$$
In order to find the three inequalities that satisfy the majorization relation $\lambda_{\psi ⊗\omega_{1}}≺\sum_{j=1}^{2}q_{j}\lambda_{\phi_{j}⊗\omega_{2}}$—notice that the fourth equality is trivial—we need to sort the elements in Equation (Equation 4) in decreasing order and denote its elements by $a^{\left(1\right)}⩾a^{\left(2\right)}⩾a^{\left(3\right)}⩾a^{\left(4\right)},b_{1}^{\left(1\right)}⩾b_{1}^{\left(2\right)}⩾b_{1}^{\left(3\right)}⩾b_{1}^{\left(4\right)}$ and $b_{2}^{\left(1\right)}⩾b_{2}^{\left(2\right)}⩾b_{2}^{\left(3\right)}⩾b_{2}^{\left(4\right)}$. It is obvious that $a^{\left(1\right)}=\alpha c,a^{\left(4\right)}=\left(1-\alpha \right)\left(1-c\right),b_{1}^{\left(1\right)}=\beta_{1}d,b_{1}^{\left(4\right)}=\left(1-\beta_{1}\right)\left(1-d\right),b_{2}^{\left(1\right)}=\beta_{2}d,b_{2}^{\left(4\right)}=\left(1-\beta_{2}\right)\left(1-d\right)$.Second, we can obtain the first and third inequalities of the majorization relation
(5)$$a^{\left(1\right)}⩽qb_{1}^{\left(1\right)}+\left(1-q\right)b_{2}^{\left(1\right)}\;\;\Leftrightarrow \;\;\alpha c⩽q\beta_{1}d+\left(1-q\right)\beta_{2}d=\beta d\Leftrightarrow d⩾\frac{\alpha c}{\beta }$$
and
(6)$$\begin{array}{cc} \sum_{i=1}^{3}a^{\left(i\right)}⩽q\sum_{i=1}^{3}b_{1}^{\left(i\right)}+\left(1-q\right)\sum_{i=1}^{3}b_{2}^{\left(i\right)}\Leftrightarrow a^{\left(4\right)}⩾qb_{1}^{\left(4\right)}+\left(1-q\right)b_{2}^{\left(4\right)}\Leftrightarrow \left(1-\alpha \right)\left(1-c\right) & \\ ⩾q\left(1-\beta_{1}\right)\left(1-d\right)+\left(1-q\right)\left(1-\beta_{2}\right)\left(1-d\right)=\left(1-\beta \right)\left(1-d\right)\Leftrightarrow d⩾\frac{\left(1-\alpha \right)c+\alpha -\beta }{1-\beta }. & \end{array}$$Third, we need to determine the next-largest element of $B_{1},B_{2}$ in Equation (Equation 4). If $\beta_{j}⩽d$, we have $\beta_{j}\left(1-d\right)⩽\left(1-\beta_{j}\right)d$; if $\beta_{j}>d$, we have $\beta_{j}\left(1-d\right)>\left(1-\beta_{j}\right)d$, j=1,2. Then, the following four cases can determine the second- and the third-largest elements in $B_{1},B_{2}$: (i) $\beta_{1}⩽d,\beta_{2}⩽d$, (ii) $\beta_{1}>d,\beta_{2}>d$, (iii) $\beta_{1}⩽d,\beta_{2}>d$ and (iv) $\beta_{1}>d,\beta_{2}⩽d$. After finding all the possibilities for $B_{1}$ and $B_{2}$, we can calculate $\sum_{i=1}^{l}b_{1}^{\left(i\right)}$ and $\sum_{i=1}^{l}b_{2}^{\left(i\right)}$ for all l=1,…,4.Finally, we still need to calculate $\sum_{i=1}^{l}a^{\left(i\right)}$ to obtain the second inequality of the majorization relation. In Equation (Equation 4), $a^{\left(1\right)}+a^{\left(2\right)}=\alpha c+\alpha \left(1-c\right)=\alpha$ or $a^{\left(1\right)}+a^{\left(2\right)}=\alpha c+\left(1-\alpha \right)c=c$; by Lemma 3, we have $a^{\left(1\right)}+a^{\left(2\right)}=\max\left\{\alpha ,c\right\}$. Based on the above results, we can obtain the second inequality of the majorization relation.(i) $\beta_{1}⩽d,\beta_{2}⩽d$: We have $b_{1}^{\left(2\right)}=\left(1-\beta_{1}\right)d$, $b_{2}^{\left(2\right)}=\left(1-\beta_{2}\right)d$, then $\sum_{i=1}^{2}a^{\left(i\right)}⩽q\sum_{i=1}^{2}b_{1}^{\left(i\right)}+\left(1-q\right)\sum_{i=1}^{2}b_{2}^{\left(i\right)}\Leftrightarrow \max\left\{\alpha ,c\right\}⩽qd+\left(1-q\right)d=d$. This case contradicts Equation (Equation 3).(ii) $\beta_{1}>d,\beta_{2}>d$: We have $b_{1}^{\left(2\right)}=\beta_{1}\left(1-d\right),b_{2}^{\left(2\right)}=\beta_{2}\left(1-d\right)$; then $\sum_{i=1}^{2}a^{\left(i\right)}⩽q\sum_{i=1}^{2}b_{1}^{\left(i\right)}+\left(1-q\right)\sum_{i=1}^{2}b_{2}^{\left(i\right)}\Leftrightarrow \max\left\{\alpha ,c\right\}⩽q\beta_{1}+\left(1-q\right)\beta_{2}=\beta$. The condition α⩽β is implied in Equation (Equation 2). In this case we have
(7)$$c⩽\beta ,d<\min\left\{\beta_{1},\beta_{2}\right\}.$$(iii) $\beta_{1}⩽d,\beta_{2}>d$: We have $b_{1}^{\left(2\right)}=\left(1-\beta_{1}\right)d,b_{2}^{\left(2\right)}=\beta_{2}\left(1-d\right)$; then $\sum_{i=1}^{2}a^{\left(i\right)}⩽q\sum_{i=1}^{2}b_{1}^{\left(i\right)}+\left(1-q\right)\sum_{i=1}^{2}b_{2}^{\left(i\right)}\Leftrightarrow \max\left\{\alpha ,c\right\}⩽qd+\left(1-q\right)\beta_{2}\Leftrightarrow \max\left\{\frac{\alpha -\left(1-q\right)\beta_{2}}{q},\frac{c-\left(1-q\right)\beta_{2}}{q}\right\}$⩽d. So
$$\max\left\{\frac{\alpha -\left(1-q\right)\beta_{2}}{q},\frac{c-\left(1-q\right)\beta_{2}}{q},\beta_{1}\right\}⩽d<\beta_{2}.$$
Since α⩽β, we have $\beta_{1}⩾\frac{\alpha -\left(1-q\right)\beta_{2}}{q}$. At the same time, we have $c<\beta_{2}$. Otherwise $c⩾\beta_{2}>qd+\left(1-q\right)\beta_{2}$; this is in contradiction with the above second inequality $\left(c⩽qd+\left(1-q\right)\beta_{2}\right)$. Combining the two conditions $c<\beta_{2}$ and $\beta_{1}<\beta <\beta_{2}$, we can divide the system into two parts: c⩽β and $\beta <c<\beta_{2}$. Similarly, for c⩽β, we have $\beta_{1}⩾\frac{c-\left(1-q\right)\beta_{2}}{q}$; for $\beta <c<\beta_{2}$, we have $\beta_{1}<\frac{c-\left(1-q\right)\beta_{2}}{q}$. This means that the above inequality can be simplified. In this case, we have
$$c⩽\beta ,\beta_{1}⩽d<\beta_{2};$$
(8)$$\beta <c<\beta_{2},\frac{c-\left(1-q\right)\beta_{2}}{q}⩽d<\beta_{2}.$$(iv) $\beta_{1}>d,\beta_{2}⩽d$: We have $b_{1}^{\left(2\right)}=\beta_{1}\left(1-d\right),b_{2}^{\left(2\right)}=\left(1-\beta_{2}\right)d$; then $\sum_{i=1}^{2}a^{\left(i\right)}⩽q\sum_{i=1}^{2}b_{1}^{\left(i\right)}+\left(1-q\right)\sum_{i=1}^{2}b_{2}^{\left(i\right)}\Leftrightarrow \max\left\{\alpha ,c\right\}⩽q\beta_{1}+\left(1-q\right)d\Leftrightarrow \max\left\{\frac{\alpha -q\beta_{1}}{1-q},\frac{c-q\beta_{1}}{1-q}\right\}⩽d$. So
$$\max\left\{\frac{\alpha -q\beta_{1}}{1-q},\frac{c-q\beta_{1}}{1-q},\beta_{2}\right\}⩽d<\beta_{1}.$$
Same as in case (iii), in this case we have
$$c⩽\beta ,\beta_{2}⩽d<\beta_{1};$$
(9)$$\beta <c<\beta_{1},\frac{c-q\beta_{1}}{1-q}⩽d<\beta_{1}.$$Combining Equations (7)–(9), we obtain the second inequality of majorization relation
$$c⩽\beta ,d<\max\left\{\beta_{1},\beta_{2}\right\};$$
(10)$$\beta <c<\max\left\{\beta_{1},\beta_{2}\right\},\min\left\{\frac{c-\left(1-q\right)\beta_{2}}{q},\frac{c-q\beta_{1}}{1-q}\right\}⩽d<\max\left\{\beta_{1},\beta_{2}\right\}.$$To sum up, combining Equations (3), (5), (6) and (10), we can obtain
$$\frac{1}{2}<c⩽\beta ,\max\left\{\frac{1}{2},\frac{\alpha c}{\beta },\frac{\left(1-\alpha \right)c+\alpha -\beta }{1-\beta }\right\}⩽d<c;$$
$$\beta <c<\max\left\{\beta_{1},\beta_{2}\right\},\max\left\{\frac{1}{2},\frac{\alpha c}{\beta },\frac{\left(1-\alpha \right)c+\alpha -\beta }{1-\beta },\min\left\{\frac{c-\left(1-q\right)\beta_{2}}{q},\frac{c-q\beta_{1}}{1-q}\right\}\right\}⩽d<c.$$
When $\frac{1}{2}<c⩽\beta$, it follows that $\frac{\alpha c}{\beta }⩾\frac{\left(1-\alpha \right)c+\alpha -\beta }{1-\beta }$. When $\beta <c<\max\left\{\beta_{1},\beta_{2}\right\}$, it follows that $\frac{\alpha c}{\beta }⩽\frac{\left(1-\alpha \right)c+\alpha -\beta }{1-\beta }$ and $\frac{1}{2}<\frac{\left(1-\alpha \right)c+\alpha -\beta }{1-\beta }$. So we obtain
$$\frac{1}{2}<c⩽\beta ,\max\left\{\frac{1}{2},\frac{\alpha c}{\beta }\right\}⩽d<c;$$
$$\beta <c<\max\left\{\beta_{1},\beta_{2}\right\},\max\left\{\frac{\left(1-\alpha \right)c+\alpha -\beta }{1-\beta },\min\left\{\frac{c-\left(1-q\right)\beta_{2}}{q},\frac{c-q\beta_{1}}{1-q}\right\}\right\}⩽d<c.$$
Thus, the proof of the proposition is completed.    □

Notice that the condition $c<\max\left\{\beta_{1},\beta_{2}\right\}$ of Proposition 1 is crucial in the recovery scheme. The condition indicates that if we want to recover the coherence loss in the above state transformation, the initial auxiliary coherent state $\omega_{1}$ must have enough coherence.

Proposition 1 tells us that the partial recovery of coherence loss in qubit state transformation is always possible by choosing appropriate qubit auxiliary coherent states. At the same time, Proposition 1 is constructive: it provides us a way to find all auxiliary coherent states in the above recovery scheme. Here, we give a specific example to explain the effectiveness of the above proposition. As in Example 1, let $\psi =|\psi \rangle \langle \psi |,\sigma =\frac{2}{5}|\phi_{1}\rangle \langle \phi_{1}|+\frac{3}{5}|\phi_{2}\rangle \langle \phi_{2}|,$ where $|\psi \rangle =\sqrt{0.63}|0\rangle +\sqrt{0.37}\left|1\rangle ,\right|\phi_{1}\rangle =\sqrt{0.6}|0\rangle +\sqrt{0.4}\left|1\rangle ,\right|\phi_{2}\rangle =\sqrt{0.8}|0\rangle +\sqrt{0.2}|1\rangle$; by Lemma 2, we can obtain $\psi \overset{IO}{\to }\sigma$. Let the auxiliary coherent state $|\omega_{1}\rangle =\sqrt{0.64}|0\rangle +\sqrt{0.36}|1\rangle$; we have c=0.64<β=0.4×0.6+0.6×0.8=0.72; according to Proposition 1, $\max\left\{\frac{1}{2},\frac{\alpha c}{\beta }\right\}=\max\left\{0.5,0.56\right\}=0.56$, so we find all auxiliary coherent states $|\omega_{2}\rangle =\sqrt{d}|0\rangle +\sqrt{1-d}|1\rangle$ such that $\psi ⊗\omega_{1}\overset{IO}{\to }\sigma ⊗\omega_{2}$ and $C_{r}\left(\omega_{2}\right)>C_{r}\left(\omega_{1}\right)$, where 0.56⩽d<0.64.

For given states $\left(\psi ,\sigma ,\omega_{1}\right)$ that satisfy $\psi \overset{IO}{\to }\sigma ,$ we find all auxiliary coherent states $\omega_{2}$ that can recover the coherence loss in a qubit system. We ask what kind of $\omega_{2}$ can maximize the recovery of coherence loss. Since $\omega_{1}$ is a given state, we have that $C_{r}\left(\omega_{1}\right)$ is a fixed constant and denote it by m, so $\Delta =C_{r}\left(\omega_{2}\right)-C_{r}\left(\omega_{1}\right)=-d\log d-\left(1-d\right)\log\left(1-d\right)-m$, where Δ is a decreasing function in $\frac{1}{2}⩽d⩽1$. That is, in order to obtain the maximum recovery of coherence loss, we only need to find the smallest d among all the possibilities of $\omega_{2}$. As in Example 1, we obtain 0.56⩽d<0.64; when we choose the final auxiliary coherent state as $|\omega_{2}\rangle =\sqrt{0.56}|0\rangle +\sqrt{0.44}|1\rangle$, we can obtain the maximum recovery of coherence loss $\Delta =C_{r}\left(\omega_{2}\right)-C_{r}\left(\omega_{1}\right)\approx 0.99-0.94=0.05$. Now we ask what happens if $\omega_{1}$ is not a given state, i.e., for given states $\left(\psi ,\sigma \right)$ that satisfy $\psi \overset{IO}{\to }\sigma$, we ask what kind of $\omega_{1}$ and $\omega_{2}$ can maximize the recovery of coherence loss. We discuss the problem in the following examples. As in Example 1, let $\psi =|\psi \rangle \langle \psi |,\sigma =\frac{2}{5}|\phi_{1}\rangle \langle \phi_{1}|+\frac{3}{5}|\phi_{2}\rangle \langle \phi_{2}|,$ where $|\psi \rangle =\sqrt{0.63}|0\rangle +\sqrt{0.37}\left|1\rangle ,\right|\phi_{1}\rangle =\sqrt{0.6}|0\rangle +\sqrt{0.4}\left|1\rangle ,\right|\phi_{2}\rangle =\sqrt{0.8}|0\rangle +\sqrt{0.2}|1\rangle$; by Lemma 2, we can obtain $\psi \overset{IO}{\to }\sigma$. According to Proposition 1, we can obtain all the possible auxiliary coherent states $\omega_{1}$ and $\omega_{2}$: (1) 0.5<c⩽0.57,0.5⩽d<c, (2) 0.57<c⩽0.72,0.875c⩽d<c, (3) 0.72<c<0.745,1.32c−0.32⩽d<c and (4) 0.745<c<0.8,2.5c−1.2⩽d<c. In Figure 1, we can see that at c=0.745,d=0.664, the recovery of coherence loss is largest, and $\Delta_{\max}=0.10183$.

**Example 2.** 
*Let $\psi =|\psi \rangle \langle \psi |,\sigma =\frac{2}{5}|\phi_{1}\rangle \langle \phi_{1}|+\frac{3}{5}|\phi_{2}\rangle \langle \phi_{2}|,$ where $|\psi \rangle =\sqrt{0.6}|0\rangle +\sqrt{0.4}|1\rangle ,$
$|\phi_{1}\rangle =\sqrt{0.63}|0\rangle +\sqrt{0.37}|1\rangle ,$
$|\phi_{2}\rangle =\sqrt{0.83}|0\rangle +\sqrt{0.17}|1\rangle$; by Lemma 2, we can obtain $\psi \overset{IO}{\to }\sigma$. According to Proposition 1, we can obtain all the possible auxiliary coherent states $\omega_{1}$ and $\omega_{2}$: (1) 0.5<c⩽0.625,0.5⩽d<c, (2) 0.625<c⩽0.75,0.8c⩽d<c and (3) 0.75<c<0.83,2.5c−1.245⩽d<c. In Figure 2, we can see that at c=0.75,d=0.6, the recovery of coherence loss is largest, and $\Delta_{\max}=0.15967$.*


**Example 3.** 
*Let $\psi =|\psi \rangle \langle \psi |,\sigma =\frac{4}{5}|\phi_{1}\rangle \langle \phi_{1}|+\frac{1}{5}|\phi_{2}\rangle \langle \phi_{2}|,$ where $|\psi \rangle =\sqrt{0.64}|0\rangle +\sqrt{0.36}|1\rangle ,$$|\phi_{1}\rangle =\sqrt{0.85}|0\rangle +\sqrt{0.15}\left|1\rangle ,\right|\phi_{2}\rangle =\sqrt{0.6}|0\rangle +\sqrt{0.4}|1\rangle$; by Lemma 2, we can obtain $\psi \overset{IO}{\to }\sigma$. According to Proposition 1, we can obtain all the possible auxiliary coherent states $\omega_{1}$ and $\omega_{2}$: (1) 0.5<c⩽0.625,0.5⩽d<c, (2) 0.625<c⩽0.8,0.8c⩽d<c, (3) 0.8<c⩽0.8125,1.8c−0.8⩽d<c and (4) 0.8125<c<0.85,5c−3.4⩽d<c. In Figure 3, we can see that at c=0.8125,d=0.6625, the recovery of coherence loss is largest, and $\Delta_{\max}=0.22619$.*


## 4. The Obtainment of Maximally Coherent State

A direct application of the above recovery scheme in a qubit system is that we can obtain the maximally coherent state under joint incoherent operations, i.e., $\psi ⊗\omega \overset{IO}{\to }\sigma ⊗\Phi_{2}$, where $|\Phi_{2}\rangle =\frac{1}{\sqrt{2}}\sum_{i=1}^{2}|i\rangle$ is a two-dimensional maximally coherent state. Examples include such main coherent states ψ=|ψ〉〈ψ|, $\sigma =\frac{2}{5}|\phi_{1}\rangle \langle \phi_{1}|+\frac{2}{5}|\phi_{2}\rangle \langle \phi_{2}|+\frac{1}{5}|\phi_{3}\rangle \langle \phi_{3}|$, where $|\psi \rangle =\sqrt{0.54}|0\rangle +\sqrt{0.46}|1\rangle ,$
$|\phi_{1}\rangle =\sqrt{0.82}|0\rangle +\sqrt{0.18}|1\rangle ,$
$|\phi_{2}\rangle =\sqrt{0.88}|0\rangle +\sqrt{0.12}\left|1\rangle ,\right|\phi_{3}\rangle =\sqrt{0.65}|0\rangle +\sqrt{0.35}|1\rangle .$ The above states satisfy the majorization relation $\lambda_{\psi }≺\sum_{j=1}^{3}q_{j}\lambda_{\phi_{j}}$, i.e., $\left(0.54,0.46\right)≺\left(0.81,0.19\right)$, so we can obtain $\psi \overset{IO}{\to }\sigma$. At the same time, there exists auxiliary coherent state $|\omega \rangle =\sqrt{0.7}|0\rangle +\sqrt{0.3}|1\rangle$ such that $\lambda_{\psi ⊗\omega }≺\sum_{j=1}^{3}q_{j}\lambda_{\phi_{j}⊗\Phi_{2}},$ i.e., $\left(0.378,0.322,0.162,0.138\right)≺\left(0.405,0.405,0.095,0.095\right);$ then, we can obtain $\psi ⊗\omega \overset{IO}{\to }\sigma ⊗\Phi_{2}$. A natural problem is how to obtain the maximally coherent state. We give a full characterization of obtaining the maximally coherent state in a qubit system in the following proposition.

**Proposition 2.** 
*For given main coherent states ψ=|ψ〉〈ψ| and $\sigma =\sum_{j=1}^{r}q_{j}|\phi_{j}\rangle \langle \phi_{j}|$, where $|\psi \rangle =\sqrt{\alpha }|0\rangle +\sqrt{1-\alpha }\left|1\rangle ,\right|\phi_{j}\rangle =\sqrt{\beta_{j}}|0\rangle +\sqrt{1-\beta_{j}}|1\rangle$, j=1,…,r, suppose $\psi \overset{IO}{\to }\sigma$; there exists an auxiliary coherent state $|\omega \rangle =\sqrt{c}|0\rangle +\sqrt{1-c}|1\rangle$ such that $\psi ⊗\omega \overset{IO}{\to }\sigma ⊗\Phi_{2}$ if and only if $\frac{1}{2}⩽\alpha ⩽\beta ⩽1$ and $\frac{1}{2}\leq c\leq \frac{\beta }{2\alpha }$, where $\beta =\sum_{j=1}^{r}q_{j}\beta_{j}$.*


**Proof.** The condition $\psi \overset{IO}{\to }\sigma$ means
(11)$$\frac{1}{2}⩽\alpha ⩽\sum_{j=1}^{r}q_{j}\beta_{j}=\beta ⩽1.$$
By Lemma 2, the proof of the proposition can be reduced to finding the condition that satisfies majorization relation $\lambda_{\psi ⊗\omega }≺\sum_{j=1}^{r}q_{j}\lambda_{\phi_{j}⊗\Phi_{2}}$ with assumption (11).First, we can obtain the squared coefficients of |ψ〉⊗|ω〉,
(12)$$A=\left\{\alpha c,\alpha \left(1-c\right),\left(1-\alpha \right)c,\left(1-\alpha \right)\left(1-c\right)\right\}.$$
Notice that for $|\phi_{j}\rangle ⊗|\Phi_{2}\rangle$, there only exist two different elements $\frac{1}{2}\beta_{j}$ and $\frac{1}{2}\left(1-\beta_{j}\right)$, where $\frac{1}{2}\beta_{j}⩾\frac{1}{2}\left(1-\beta_{j}\right)$, j=1,…,r. For convenience, we denote the non-increasing coefficients of $|\phi_{j}\rangle ⊗|\Phi_{2}\rangle$ by $b_{j}^{\left(1\right)}⩾b_{j}^{\left(2\right)}⩾b_{j}^{\left(3\right)}⩾b_{j}^{\left(4\right)},j=1,\ldots ,r$. Similarly, we need to sort the elements in A in decreasing order and denote its elements by $a^{\left(1\right)}⩾a^{\left(2\right)}⩾a^{\left(3\right)}⩾a^{\left(4\right)}$. It is obvious that $a^{\left(1\right)}=\alpha c,a^{\left(4\right)}=\left(1-\alpha \right)\left(1-c\right)$.Second, we can obtain the first and third inequalities of the majorization relation
(13)$$\begin{array}{c} a^{\left(1\right)}⩽\sum_{j=1}^{r}q_{j}b_{j}^{\left(1\right)}\Leftrightarrow \alpha c⩽\frac{1}{2}\sum_{j=1}^{r}q_{j}\beta_{j}=\frac{\beta }{2}\Leftrightarrow c⩽\frac{\beta }{2\alpha } \end{array}$$
and
(14)$$\begin{array}{cc} \sum_{i=1}^{3}a^{\left(i\right)}⩽\sum_{j=1}^{r}\sum_{i=1}^{3}q_{j}b_{j}^{\left(i\right)}\Leftrightarrow a^{\left(4\right)}⩾\sum_{j=1}^{r}q_{j}b_{j}^{\left(4\right)}\Leftrightarrow & \\ \left(1-\alpha \right)\left(1-c\right)⩾\frac{1}{2}\sum_{j=1}^{r}q_{j}\left(1-\beta_{j}\right)\Leftrightarrow c⩽\frac{1-2\alpha +\beta }{2\left(1-\alpha \right)}. \end{array}$$Finally, we still need to calculate $\sum_{i=1}^{l}a^{\left(i\right)}$ to obtain the second inequality of the majorization relation. In Equation (Equation 12), $a^{\left(1\right)}+a^{\left(2\right)}=\alpha c+\alpha \left(1-c\right)=\alpha$ or $a^{\left(1\right)}+a^{\left(2\right)}=\alpha c+\left(1-\alpha \right)c=c$; by Lemma 3, we have $a^{\left(1\right)}+a^{\left(2\right)}=\max\left\{\alpha ,c\right\}$. Then the second inequality of the majorization relation can be written as $\max\left\{\alpha ,c\right\}⩽\sum_{j=1}^{r}q_{j}\beta_{j}=\beta .$ The condition α⩽β is implied in Equation (Equation 11). In this case, we have
(15)c⩽β.To sum up, combining Equations (13)–(15) and $\frac{1}{2}\leq c\leq 1$, we can obtain
$$\frac{1}{2}\leq c⩽\min\left\{\frac{\beta }{2\alpha },\frac{1-2\alpha +\beta }{2\left(1-\alpha \right)},\beta \right\}.$$
Notice that $\frac{1}{2}⩽\alpha ⩽\beta ⩽1$; we have $\beta ⩾\frac{\beta }{2\alpha }$ and $\frac{1-2\alpha +\beta }{2\left(1-\alpha \right)}⩾\frac{\beta }{2\alpha }$. So the above inequality can be simplified as
$$\frac{1}{2}\leq c⩽\frac{\beta }{2\alpha }.$$
Thus, the proof of the proposition is completed.    □

The above proposition gives us a new way to obtain the maximally coherent state. For the transformation $\psi \overset{IO}{\to }\sigma$, as long as the parameter *c* of the auxiliary coherent state satisfies $\frac{1}{2}\leq c⩽\frac{\beta }{2\alpha }$, we can prepare the maximally coherent state under joint incoherent operations.

## 5. Coherence-Assisted Transformation for Arbitrary Finite-Dimensional Main Coherent States and Low-Dimensional Auxiliary Coherent States

In Section 3, we obtain that the partial recovery of coherence loss in two-dimensional state transformations is always possible by choosing appropriate two-dimensional auxiliary coherent states. In fact, we can show that the coherence-assisted transformation for arbitrary finite dimensional main coherent states and two-dimensional auxiliary coherent states is always possible, and the coherence loss can be partially recovered simultaneously. Specifically, for given main coherent states ψ=|ψ〉〈ψ| and $\sigma =\sum_{j=1}^{r}q_{j}|\phi_{j}\rangle \langle \phi_{j}|$ that satisfy $\psi \overset{IO}{\to }\sigma$, where $|\psi \rangle =\sum_{i=1}^{n}\sqrt{\alpha_{i}}|i\rangle ,|\phi_{j}\rangle =\sum_{i=1}^{n}\sqrt{\beta_{ji}}|i\rangle ,j=1,\ldots ,r$, let $\lambda_{\psi }=\left(\alpha_{1},\ldots ,\alpha_{n}\right),\lambda_{\phi_{j}}=\left(\phi_{j1},\ldots ,\phi_{jn}\right)$; then, the squared coefficients of the above states satisfy the majorization relation $\lambda_{\psi }≺\sum_{j=1}^{r}q_{j}\lambda_{\phi_{j}}$. If all inequalities in above majorization relation are strict, we call it a strict majorization relation and denote it by $\lambda_{\psi }⪵\sum_{j=1}^{r}q_{j}\lambda_{\phi_{j}}$. Next, we show that there exist two-dimensional auxiliary coherent states such that the recovery is always possible in this case.

**Proposition 3.** 
*If $\lambda_{\psi }⪵\sum_{j=1}^{r}q_{j}\lambda_{\phi_{j}}$, then there exist two-dimensional auxiliary coherent pure states $\omega_{1}$ and $\omega_{2}$ such that $\psi ⊗\omega_{1}\overset{IO}{\to }\sigma ⊗\omega_{2}$ and $C_{r}\left(\omega_{2}\right)>C_{r}\left(\omega_{1}\right)$.*


**Proof.** The condition $\lambda_{\psi }⪵\sum_{j=1}^{r}q_{j}\lambda_{\phi_{j}}$ means $\psi \overset{IO}{\to }\sigma$. Let $|\omega \left(p\right)\rangle$ be a two-dimensional state with $\lambda_{\omega \left(p\right)}=\left(p,1-p\right)$; then for any $p\in \left(\frac{1}{2},1\right)$, we have $\psi ⊗\omega \left(p\right)\overset{IO}{\to }\sigma ⊗\omega \left(p\right)$. Notice that the selection of p determines the orders of the squared coefficients of $|\psi \rangle ⊗|\omega \left(p\right)\rangle$ and $|\phi_{j}\rangle ⊗|\omega \left(p\right)\rangle$, j=1,…r. Instead, let us start from the perspective of the orders of the squared coefficients. In fact, there is only a finite number of possible individual orders (at most n! orders) of the squared coefficients of $|\psi \rangle ⊗|\omega \left(p\right)\rangle$ and $|\phi_{j}\rangle ⊗|\omega \left(p\right)\rangle$, j=1,…r. At the same time, there is only a finite number of possible whole orders (at most ${\left(n!\right)}^{r}$ orders) of the squared coefficients of $|\phi_{j}\rangle ⊗|\omega \left(p\right)\rangle$, j=1,…r. For each of the above (whole) orders, we can obtain a feasible set of *p*, i.e., the order is valid in the feasible set of *p*. Each nonempty feasible set is either a discrete point or an interval in $\left(\frac{1}{2},1\right)$. Furthermore, there are an infinite number of $p\in \left(\frac{1}{2},1\right)$ and a finite number of the orders of the squared coefficients such that $\lambda_{\psi ⊗\omega \left(p\right)}≺\sum_{j=1}^{r}q_{j}\lambda_{\phi_{j}⊗\omega \left(p\right)}$; then there exists at least one order for which the corresponding feasible set of p is an interval of non-zero length $\left(a,b\right)$, where $\frac{1}{2}<a<b<1$.Based on the above analysis, let p belong to the above nontrivial feasible set F. We are going to show that $\lambda_{\psi ⊗\omega \left(p\right)}⪵\sum_{j=1}^{r}q_{j}\lambda_{\phi_{j}⊗\omega \left(p\right)}$ for all values of p∈F, except at most 2n−1 nontrivial values of p. In the majorization relation $\lambda_{\psi ⊗\omega \left(p\right)}≺\sum_{j=1}^{r}q_{j}\lambda_{\phi_{j}⊗\omega \left(p\right)}$, if one of 2n−1 nontrivial inequalities is an equality, we have
(16)$$p\sum_{i=1}^{x}\alpha_{i}+\left(1-p\right)\sum_{i=1}^{y}\alpha_{i}=p\sum_{j=1}^{r}\sum_{i=1}^{s_{j}}q_{j}\beta_{ji}+\left(1-p\right)\sum_{j=1}^{r}\sum_{i=1}^{t_{j}}q_{j}\beta_{ji},$$
where $r\left(x+y\right)=\sum_{j=1}^{r}\left(s_{j}+t_{j}\right)$, x⩾y, $s_{j}⩾t_{j}$ for all j=1,…,r. Equivalently,
(17)$$\left(\sum_{i=1}^{x}\alpha_{i}-\sum_{i=1}^{y}\alpha_{i}-\sum_{j=1}^{r}\sum_{i=1}^{s_{j}}q_{j}\beta_{ji}+\sum_{j=1}^{r}\sum_{i=1}^{t_{j}}q_{j}\beta_{ji}\right)p=\sum_{j=1}^{r}\sum_{i=1}^{t_{j}}q_{j}\beta_{ji}-\sum_{i=1}^{y}\alpha_{i}$$
Notice that Equation (Equation 17) can be divided into two cases: (i) Equation (Equation 17) determines a value of *p* or (ii) Equation (Equation 17) is independent of the value of *p*—it does not determine a value of *p*. In fact, Case (ii) is impossible because Equation (Equation 17) is independent of the value of *p* if and only if $\sum_{i=1}^{x}\alpha_{i}=\sum_{j=1}^{r}\sum_{i=1}^{s_{j}}q_{j}\beta_{ji}$ and $\sum_{i=1}^{y}\alpha_{i}=\sum_{j=1}^{r}\sum_{i=1}^{t_{j}}q_{j}\beta_{ji}$. As a result of $\lambda_{\psi }⪵\sum_{j=1}^{r}q_{j}\lambda_{\phi_{j}}$, we have $rx>\sum_{j=1}^{r}s_{j}$ and $ry>\sum_{j=1}^{r}t_{j}$, which is in contradiction with the condition $r\left(x+y\right)=\sum_{j=1}^{r}\left(s_{j}+t_{j}\right)$. Therefore, each nontrivial equation in the majorization relation $\lambda_{\psi ⊗\omega \left(p\right)}≺\sum_{j=1}^{r}q_{j}\lambda_{\phi_{j}⊗\omega \left(p\right)}$ corresponds to a fixed value of *p*. Moreover, there are at most 2n−1 nontrivial equalities, i.e., there are at most 2n−1 discrete values of p such that $\lambda_{\psi ⊗\omega \left(p\right)}≺\sum_{j=1}^{r}q_{j}\lambda_{\phi_{j}⊗\omega \left(p\right)}$ is not strict. Thus, we show that $\lambda_{\psi ⊗\omega \left(p\right)}⪵\sum_{j=1}^{r}q_{j}\lambda_{\phi_{j}⊗\omega \left(p\right)}$ for all values of p∈F, except at most 2n−1 nontrivial values of *p*. That is to say, for such values of *p*, the majorization relation is strict and the order of the squared coefficients is preserved.Thus, there exists a $p\in F\subseteq \left(\frac{1}{2},1\right)$ and a $0<\varepsilon <\frac{1}{2}$ such that $\lambda_{\psi ⊗\omega \left(p\right)}≺\sum_{j=1}^{r}q_{j}\lambda_{\phi_{j}⊗\omega \left(p-\varepsilon \right)}$. Let $|\omega_{1}\rangle =|\omega \left(p\right)\rangle ,|\omega_{2}\rangle =|\omega \left(p-\varepsilon \right)\rangle$; it obvious that $\psi ⊗\omega_{1}\overset{IO}{\to }\sigma ⊗\omega_{2}$ and $C_{r}\left(\omega_{2}\right)>C_{r}\left(\omega_{1}\right)$. Thus, the proof of the proposition is completed.    □

The basic idea involved in the proof is as follows. First, we select a two-dimensional auxiliary coherent state $|\omega \left(p\right)\rangle$ with $\lambda_{\omega \left(p\right)}=\left(p,1-p\right)$; since $\psi \overset{IO}{\to }\sigma$, then for any $p\in \left(\frac{1}{2},1\right)$ we have $\psi ⊗\omega \left(p\right)\overset{IO}{\to }\sigma ⊗\omega \left(p\right)$. Second, we show that the majorization relation is strict for all values of p∈F, except at most 2n−1 nontrivial values of p. This allows a perturbation of *p* to $p-\varepsilon \left(\varepsilon >0\right)$ on the right side of the 2n inequalities such that the inequalities are still satisfied after perturbation. It also allows the order of the squared coefficients to be preserved after perturbation. Finally, the above two facts reveal the existence of ε, i.e., there exists a $0<\varepsilon <\frac{1}{2}$ such that $\lambda_{\psi ⊗\omega \left(p\right)}≺\sum_{j=1}^{r}q_{j}\lambda_{\phi_{j}⊗\omega \left(p-\varepsilon \right)}$. Let $|\omega_{1}\rangle =|\omega \left(p\right)\rangle ,|\omega_{2}\rangle =|\omega \left(p-\varepsilon \right)\rangle$; it is obvious that $\psi ⊗\omega_{1}\overset{IO}{\to }\sigma ⊗\omega_{2}$ and $C_{r}\left(\omega_{2}\right)>C_{r}\left(\omega_{1}\right)$. For example, for the main coherent states $\psi =|\psi \rangle \langle \psi |,\sigma =\frac{1}{4}|\phi_{1}\rangle \langle \phi_{1}|+\frac{3}{4}|\phi_{2}\rangle \langle \phi_{2}|,$ where $|\psi \rangle =\sqrt{0.4}|0\rangle +\sqrt{0.3}|1\rangle +\sqrt{0.2}|2\rangle +\sqrt{0.1}|3\rangle ,$
$|\phi_{1}\rangle =\sqrt{0.6}|0\rangle +\sqrt{0.2}|1\rangle +\sqrt{0.2}|2\rangle ,$
$|\phi_{2}\rangle =\sqrt{0.5}|0\rangle +\sqrt{0.3}|1\rangle +\sqrt{0.2}|2\rangle$, by Lemma 2, we can obtain $\psi \overset{IO}{\to }\sigma$. Choosing p=0.8, then we have $\lambda_{\psi ⊗\omega \left(p\right)}≺\sum_{j=1}^{r}q_{j}\lambda_{\phi_{j}⊗\omega \left(p\right)}$. At the same time, we can calculate that for 0<ε<0.08, the order of the squared coefficients is preserved and the above states satisfy the majorization relation $\lambda_{\psi ⊗\omega \left(p\right)}≺\sum_{j=1}^{r}q_{j}\lambda_{\phi_{j}⊗\omega \left(p-\varepsilon \right)}$. Let $|\omega_{1}\rangle =|\omega \left(0.8\right)\rangle ,|\omega_{2}\rangle =|\omega \left(0.75\right)\rangle$; we can obtain $\psi ⊗\omega_{1}\overset{IO}{\to }\sigma ⊗\omega_{2}$ and $C_{r}\left(\omega_{2}\right)>C_{r}\left(\omega_{1}\right)$.

The above proposition shows that if $\lambda_{\psi }$ is strictly majorized by $\sum_{j=1}^{r}q_{j}\lambda_{\phi_{j}}$, recovery with two-dimensional auxiliary coherent states is always possible. We now ask what happens if $\lambda_{\psi }$ is not strictly majorized by $\sum_{j=1}^{r}q_{j}\lambda_{\phi_{j}}$. Next, we show that if there exist certain equalities in the majorization relation, the recovery scheme is impossible with the help of two- or three-dimensional auxiliary coherent states.

**Proposition 4.** 
*If $\alpha_{1}=\sum_{j=1}^{r}q_{j}\beta_{j1}$ or $\alpha_{n}=\sum_{j=1}^{r}q_{j}\beta_{jn}$, then the recovery scheme is not possible with the help of two-dimensional auxiliary coherent states. Furthermore, if $\alpha_{1}=\sum_{j=1}^{r}q_{j}\beta_{j1}$ and $\alpha_{n}=\sum_{j=1}^{r}q_{j}\beta_{jn}$, then the recovery scheme is not possible with the help of three-dimensional auxiliary coherent states.*


**Proof.** (1) When $\alpha_{1}=\sum_{j=1}^{r}q_{j}\beta_{j1}$ or $\alpha_{n}=\sum_{j=1}^{r}q_{j}\beta_{jn}$, suppose there exist two-dimensional auxiliary coherent states $|\omega_{1}\rangle =\sqrt{c}|0\rangle +\sqrt{1-c}|1\rangle$ and $|\omega_{2}\rangle =\sqrt{d}|0\rangle +\sqrt{1-d}|1\rangle$ such that $\psi ⊗\omega_{1}\overset{IO}{\to }\sigma ⊗\omega_{2}$ and $C_{r}\left(\omega_{2}\right)>C_{r}\left(\omega_{1}\right)$. From $C_{r}\left(\omega_{2}\right)>C_{r}\left(\omega_{1}\right)$, we obtain c>d. From $\psi ⊗\omega_{1}\overset{IO}{\to }\sigma ⊗\omega_{2}$, we have $\alpha_{1}c⩽\sum_{j=1}^{r}q_{j}\beta_{j1}d$ and $\alpha_{n}\left(1-c\right)\geq \sum_{j=1}^{r}q_{j}\beta_{jn}\left(1-d\right)$. Therefore, if $\alpha_{1}=\sum_{j=1}^{r}q_{j}\beta_{j1}$, it follows that c⩽d. This is in contradiction with c>d; then the hypothesis is not valid. The case for $\alpha_{n}=\sum_{j=1}^{r}q_{j}\beta_{jn}$ is the same. So the recovery scheme is not possible with the help of two-dimensional auxiliary coherent states.(2) When $\alpha_{1}=\sum_{j=1}^{r}q_{j}\beta_{j1}$ and $\alpha_{n}=\sum_{j=1}^{r}q_{j}\beta_{jn}$, suppose there exist three-dimensional auxiliary coherent states $|\omega_{1}\rangle =\sqrt{c_{1}}|0\rangle +\sqrt{c_{2}}|1\rangle +\sqrt{1-c_{1}-c_{2}}|2\rangle$ and $|\omega_{2}\rangle =\sqrt{d_{1}}|0\rangle +\sqrt{d_{2}}|1\rangle +\sqrt{1-d_{1}-d_{2}}|2\rangle$ such that $\psi ⊗\omega_{1}\overset{IO}{\to }\sigma ⊗\omega_{2}$ and $C_{r}\left(\omega_{2}\right)>C_{r}\left(\omega_{1}\right)$. From the condition $\psi ⊗\omega_{1}\overset{IO}{\to }\sigma ⊗\omega_{2}$, we have $\alpha_{1}c_{1}⩽\sum_{j=1}^{r}q_{j}\beta_{j1}d_{1}$ and $\alpha_{n}\left(1-c_{1}-c_{2}\right)\geq \sum_{j=1}^{r}q_{j}\beta_{jn}\left(1-d_{1}-d_{2}\right)$. Therefore, if $\alpha_{1}=\sum_{j=1}^{r}q_{j}\beta_{j1}$ and $\alpha_{n}=\sum_{j=1}^{r}q_{j}\beta_{jn}$, it follows that $c_{1}⩽d_{1}$ and $c_{1}+c_{2}⩽d_{1}+d_{2}$. So the majorization relation $\left(c_{1},c_{2},1-c_{1}-c_{2}\right)≺\left(d_{1},d_{2},1-d_{1}-d_{2}\right)$ holds; then we can obtain $|\omega_{1}\rangle \overset{IO}{\to }|\omega_{2}\rangle$, i.e., $C_{r}\left(\omega_{2}\right)⩽C_{r}\left(\omega_{1}\right)$. This is in contradiction with $C_{r}\left(\omega_{2}\right)>C_{r}\left(\omega_{1}\right)$; then the hypothesis is not valid. So the recovery scheme is not possible with the help of three-dimensional auxiliary coherent states.    □

Thus, under certain conditions, we show that the recovery for arbitrary finite dimensional main coherent states and two-dimensional auxiliary coherent states is always possible. The above discussion has practical implications because low-dimensional coherent states are easier to prepare in the context of current techniques and tools.

## 6. Discussion

In this paper, for the transformation from a pure state to a mixed state under incoherent operations, we add auxiliary coherent states such that the transformation can still be realized under joint incoherent operations; such a process is called a ’coherence-assisted transformation’. When the coherence of an auxiliary coherent state increases, the reduced coherence of the initial main coherent state can be partially transformed to the final auxiliary coherent state, so the coherence loss can be partially recovered. We first discuss the coherence-assisted transformation for qubit states and give the sufficient and necessary condition for the partial recovery of coherence loss. The maximum of the recovery of coherence loss is also studied in this case. We also give the sufficient and necessary condition for obtaining the maximally coherent state in a qubit system. If the parameter of the initial auxiliary coherent state satisfies a certain condition, we can obtain a two-dimensional maximally coherent state. Furthermore, the coherence-assisted transformation for qubit states can be extended to the general case, i.e., arbitrary finite-dimensional main coherent states and low-dimensional auxiliary coherent states. In this case, we show that if the main coherent states satisfy a strictly majorization relation, there exist two-dimensional auxiliary coherent states that can realize the above recovery scheme. At the same, there are some open questions: What is the relation between the dimensionality of auxiliary coherent states and the amount of coherence recovery? For the transformation between two mixed states, what is the condition for the partial recovery of coherence loss? We hope that the results presented in this paper contribute to a better understanding of the resource theory of quantum coherence.

## Figures and Tables

**Figure 1 entropy-25-01375-f001:**
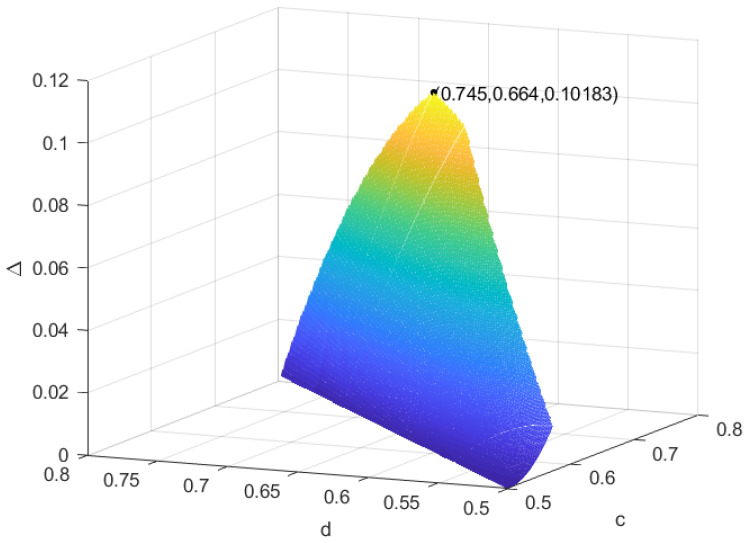
The amount of the recovery of coherence loss in Example 1.

**Figure 2 entropy-25-01375-f002:**
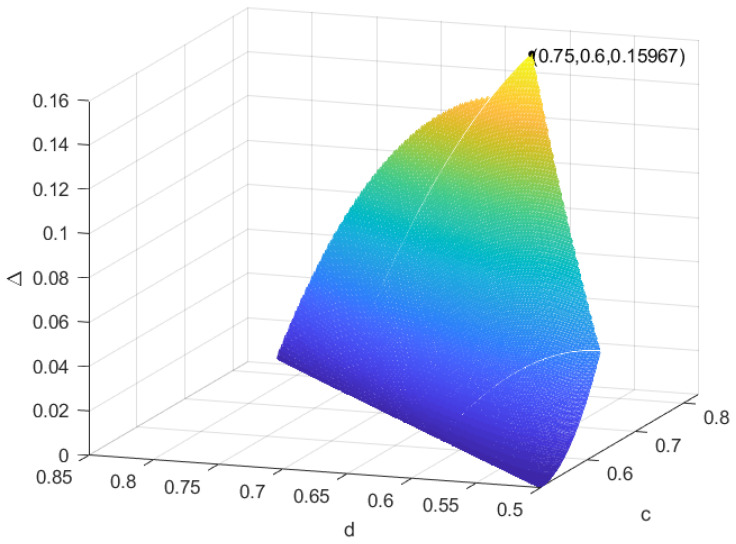
The amount of the recovery of coherence loss in Example 2.

**Figure 3 entropy-25-01375-f003:**
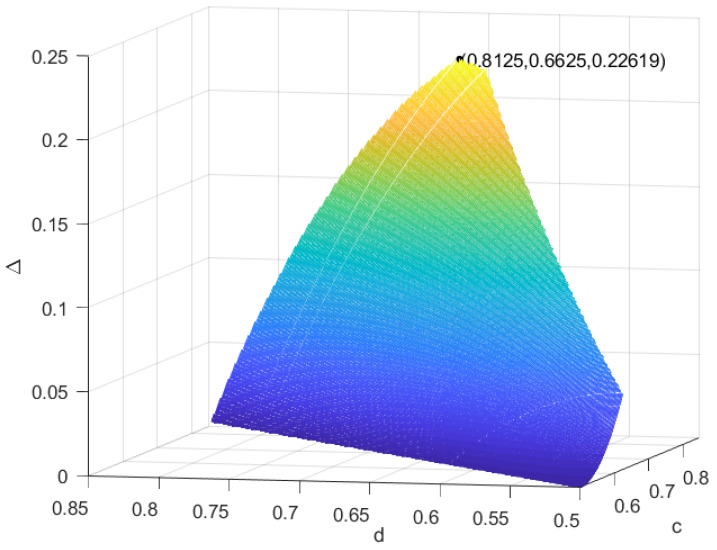
The amount of the recovery of coherence loss in Example 3.

## Data Availability

Data sharing are not applicable.

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
