# Peer review of "Partial Recovery of Coherence Loss in Coherence-Assisted Transformation"

_entropy, 2023, doi:10.3390/e25101375_

Round 1

Author Response

We have revised the paper according to the reviewers' comments.

Reviewer 2 Report

In this manuscript, the authors studied the problem of coherence recovery under the coherent-assisted transformation. A necessary and sufficient condition for partially recovering coherence in the case of qubit states is provided. In addition, the high-dimensional situation was also discussed. The result is potentially useful in quantum information and also in fundamentals of quantum mechanics. Below is my comment:

To my knowledge, there are many coherence measures, such as distance-based measures, entanglement-based measures and Skew-information-based measures, etc. Why did the authors choose to use the relative entropy measure instead of others? Is there a similar conclusion for other coherent measures? 

The introduction is well written while some important literatures are missing in the first paragraph. To enhance the impact of this work, it shall ge nice for the authors to add some related literatures, such as Nature 549, 70–73 (2017) for quantum information processing (quantum teleportation), and Phys. Rev. A 98,  062323 (2018) and Phys. Rev. A, 93, 042324 (2016) for quantum cryptography, etc.

It can be published after minor revisions related to the coments above.

Author Response

We revised the paper according to the reviewers' comments

Reviewer 3 Report

This manuscript examines coherence assisted transformation under incoherent operations from a pure state to a mixed state. The authors show that a partial recovery of coherence loss is available by using auxiliary coherent states. They provide the necessary conditions when considering qubit systems and provide the prescription for obtaining the maximally coherent state. They also extend this to consider finite dimensional systems and low dimensional auxiliary coherent states.  

The manuscript is well-written and well-presented and should be of interest to the community. I recommend publication.

There are a few typographical errors.  ‘scheme’ p. 2, ‘satisfy’ p. 3, ‘states’ p. 3, ‘suppose’ p. 4, ‘satisfy’ p. 7, ‘relation’ p. 9, ‘squared’ p. 9, ‘preserved’ p. 10 (twice).

Author Response

(The authors gave the same response as above.)

Round 2

Reviewer 1 Report

I am satisfied with the revision and the added comments.
The manuscript may be published in the present form.